# Osteopontin on the Dental Implant Surface Promotes Direct Osteogenesis in Osseointegration

**DOI:** 10.3390/ijms23031039

**Published:** 2022-01-18

**Authors:** Sanako Makishi, Tomohiko Yamazaki, Hayato Ohshima

**Affiliations:** 1Division of Anatomy and Cell Biology of the Hard Tissue, Department of Tissue Regeneration and Reconstruction, Niigata University Graduate School of Medical and Dental Sciences, Niigata 951-8514, Japan; histogirl@dent.niigata-u.ac.jp; 2Research Center for Functional Materials, National Institute for Materials Science, Tsukuba 305-0047, Japan; YAMAZAKI.Tomohiko@nims.go.jp

**Keywords:** dental implants, maxilla, mice (knockout), osseointegration, osteopontin, titanium, tooth extraction

## Abstract

After dental implantation, osteopontin (OPN) is deposited on the hydroxyapatite (HA) blasted implant surface followed by direct osteogenesis, which is significantly disturbed in *Opn*-knockout (KO) mice. However, whether applying OPN on the implant surface promotes direct osteogenesis remains unclarified. This study analyzed the effects of various OPN modified protein/peptides coatings on the healing patterns of the bone-implant interface after immediately placed implantation in the maxilla of four-week-old *Opn*-KO and wild-type (WT) mice (*n* = 96). The decalcified samples were processed for immunohistochemistry for OPN and Ki67 and tartrate-resistant acid phosphatase histochemistry. In the WT mice, the proliferative activity in the HA binding peptide-OPN mimic peptide fusion coated group was significantly higher than that in the control group from day 3 to week 1, and the rates of OPN deposition and direct osteogenesis around the implant surface significantly increased in the recombinant-mouse-OPN (rOPN) group compared to the Gly-Arg-Gly-Asp-Ser peptide group in week 2. The rOPN group achieved the same rates of direct osteogenesis and osseointegration as those in the control group in a half period (week 2). None of the implant surfaces could rescue the direct osteogenesis in the healing process in the *Opn*-KO mice. These results suggest that the rOPN coated implant enhances direct osteogenesis during osseointegration following implantation.

## 1. Introduction

The goal of hard tissue engineering is to join engineered constructs that improve the healing process of damaged tissue and restore or maintain its function [1]. Inspired by nature, scientists in tissue engineering fields have been trying to develop engineered constructs mimicking the natural wound healing process. The incorporation of mineral into hard tissues, such as bones and teeth, is essential to give them strength and structure for body support and function. An extracellular matrix (ECM) plays a role in this process not only as a structure, but also as a key regulator of the mineralization [2]. For example, after the implantation of tissue engineering scaffolds into an organism, protein adsorption to its surface occurs in a moment, mediates the cell adhesion, and also provides signals to cells. This is followed by the release of active compounds for signaling and ECM deposition by cells, cell proliferation, and cell differentiation [3]. Focusing on the extracellular environment, ECM protein is the most important biomolecule for regulating these cellular events. In dental hard tissue engineering, sophisticated strategies applying hydroxyapatite (HA) as scaffolds have been developed due to the property of this material, osteoinductivity. The substrate of HA activates the monocyte/macrophage-lineage cells followed by receptor activator of nuclear factor kappa-β (RANK) and RANK ligand (RANKL) interaction and the subsequent initiation of osteogenesis [4]. Also, HA allows the differentiation of osteoclast precursors into mature osteoclasts [5]. However, the effects of ECM protein on dental implant-associated HA scaffold have not previously come into the spotlight. Now, hard tissue engineering is rapidly developing and needs the fusion of engineering and biological aspects. Recent studies have started to represent the potential of bio-hybrid dental implants using stem cells [6,7]. Although these regenerative therapies have been used to treat tooth loss, further biological approaches based on biological findings are expected to improve dental implant therapy.

Osseointegration is defined by Brånemark as a direct contact of living bone with the surface of an implant at the light microscopic level of magnification [8]. It consists of direct osteogenesis and indirect osteogenesis: this concept is important in the healing process after dental implantation. After implantation, osteoblasts may lay down on the damaged pre-existing bone surface, leading to “indirect osteogenesis”: bone formation occurs from the bone surface. Meanwhile, some osteoblasts are recruited on the implant surface, leading to “direct osteogenesis”: bone formation occurs from the implant surface [9]. As both types of osteogenesis develop, the stability of the implant increases, which is referred to as “secondary stability.” Regarding the timing of implant placement, there are three protocols for dental implant therapies: late placement (a post-extraction healing period of at least six months before implant placement), immediate implant placement (implant placement into fresh extraction sockets immediately after extraction), and early implant placement (implant placement following complete soft tissue coverage of the extraction socket). Although there is still a controversy on which protocol is more clinically effective, a meta-analysis study revealed that there were no statistically significant differences in the implant outcomes (risk of implant failure) between early implant placement protocol and immediate or delayed implant placement protocols [10]. Additionally, it was reported that there are no significant differences in the chronological healing process at the bone-implant interface between immediate and delayed placement groups at the cellular level [11]. Acquiring secondary implant stability is important for successful implant therapy [12]. Excessive implant movement after insertion will induce a fibrous tissue development around the implant, which ultimately leads to clinical failure [13]. Therefore, promoting direct osteogenesis as well as indirect osteogenesis could contribute to achieving faster osseointegration, preventing excessive implant movement (clinical failure) in the early days after insertion. However, the mechanisms enhancing either type of osteogenesis remain to be clarified.

Osteopontin (OPN) is an ECM protein that is a prominent component in bone. Interestingly, direct osteogenesis on the dental implant surface is significantly disturbed in the *Opn*-knockout (KO) mice, while indirect osteogenesis is not affected [9]. OPN has been shown to play a role on bone mineralization, wound healing, angiogenesis, cell adhesion, cell differentiation and foreign body response [14,15], since it has several binding sites with HA crystals, collagen, various integrins through the Arg-Gly-Asp (RGD) motif, and calcium ions [14]. The RGD motif in OPN is a major factor affecting the osteoclast attachment before bone resorption [16]. Some proteins containing the RGD motif are necessary for the actin ring formation and polarization in osteoclasts [17,18]. A decellularized/demineralized bone matrix contains high concentration of OPN protein and this is one of the reasons why the matrix is osteoinductive [19]. In fact, OPN coated implants showed osteoinductive capacities histologically in the rat femur [20]. Furthermore, the recruitment of osteoclasts and OPN deposition on the dental implant surface is followed by direct osteogenesis [9,11]. Considering these unique behaviors of OPN in damaged tissue, OPN coated dental implant was chosen as our experimental material for promoting direct osteogenesis. This study aimed to analyze the effects of various OPN modified protein/peptides coating on the healing patterns of the bone-implant interface after immediately placed implantation in the maxilla of 4-week-old *Opn*-KO and wild-type (WT) mice. In this paper, we report that the recombinant-mouse-OPN (rOPN) coated HA blasted (HAB)-implants achieve the same rates of direct osteogenesis and osseointegration as those in the control group in half the period. None of the implant surfaces can rescue the direct osteogenesis in the healing process in the *Opn*-KO mice. These results suggest that the rOPN coated HAB-implant enhances direct osteogenesis during osseointegration following implantation.

## 2. Results

### 2.1. Histological Changes in the Opn-KO Mice

The inflammatory phase including infiltration of numerous inflammatory cells continued during the examined periods (until week 2) in the *Opn*-KO mice. In week 2, there was a small amount of direct osteogenesis on the implant surface (Appendix A). The *Opn*-KO mice showed the lack of OPN-immunoreactivity at the bone-implant interface (Appendix A) that was observed in all groups of the WT mice (see below).

### 2.2. On Day 5 to Week 2 in the WT Mice

On day five, the infiltration of inflammatory cells and spindle-shaped or flattened cells was observed at the bone-implant interface (Figure 1a,g,m and Appendix A) with weak OPN-positive immunoreaction at the bottom parts of threads and the cement lines of the pre-existing bone in the control, rOPN, Gly-Arg-Gly-Asp-Ser peptide (RGDS), OPN mimic peptide (OPNpep), and HA binding peptide-OPN mimic peptide fusion (HABP-OPNpep) groups (Figure 1d,j,p and Appendix A). OPN-immunoreactivity gradually became intense and elongated along the implant surface until week 2 (Figure 1e,f,k,l,q,r and Appendix A), while the formation of direct osteogenesis was clearly observed after week 1 (Figure 1b,c,h,i,n,o and Appendix A). Partially, the indirect osteogenesis progressed from the pre-existing bone in addition to the direct osteogenesis to achieve osseointegration until week 2 (Figure 2a–f). The OPN-immunoreactive lines coincided with the places where the direct osteogenesis occurred, although some area lacked the OPN reaction (Figure 2d–i). In the rOPN group, the rates of direct osteogenesis and OPN-positive perimeter around the implant surface significantly increased compared with that in the RGDS group in week 2 (Figure 2j). Moreover, OPN-positive perimeter in the rOPN group was significantly higher than that in the HABP-OPNpep group. Meanwhile, the RGDS group showed significantly decreased OPN-positive perimeter compared with the OPNpep group. The rate of osseointegration comprising direct and indirect osteogenesis was statistically analyzed in the WT mice (Figure 3). The rOPN group showed a significantly increased osseointegration rate compared with the RGDS group. There were no significant differences in the rate of direct osteogenesis between the rOPN group at week 2 and the control group at week 4.

### 2.3. Cell Proliferation in the WT Mice

Proliferative activities gradually increased and then decreased during day 3 to week 1 (Figure 4a–l). The proliferative activity in the HABP-OPNpep group was significantly higher than that in the control group from day 3 to week 1 or those in rOPN and RGDS groups from day 5 to week 1 (Figure 4m). In contrast, the rOPN group, where the increased direct osteogenesis occurred, represented no significant differences compared with other groups.

### 2.4. Tartrate-Resistant Acid Phosphatase (TRAP) Activity in the WT Mice

On day 3, the bone-implant interface lacked the intense TRAP activity, while it was observed on the outer and inner surfaces of the pre-existing bone in all groups (Appendix A). The intense TRAP activity on the implant surface was seen on day 5 (Appendix A). It gradually increased and numerous TRAP-positive cells appeared at the bone-implant interface on week 1 (Appendix A).

## 3. Discussion

This study demonstrated that rOPN protein coating on the implant surface accelerates direct osteogenesis. OPN is thought to regulate osteoclast migration, adhesion, differentiation, and activation leading to the secretion of OPN on the resorption site in bone remodeling, which also affects the osteoblast migration, adhesion, differentiation to form a bone matrix [21,22]. The OPN protein layer conditioned by osteoclasts on a HA disc also increases human osteoblast proliferation. Although the functional role of OPN in osteogenesis has not been fully understood, our in vivo study provides evidence that rOPN has a positive effect on osteoblasts in direct osteogenesis after implantation, supporting the positive effects of OPN on osteogenic cells demonstrated by a previous study [23]. In contrast, it has been long recognized that OPN has negative effects on the mineralization process, likely through inhibition of nucleation [24,25] and growth [26,27] of HA crystals. Similarly, OPN is a potent negative regulator of osteogenesis by inhibiting osteoblast proliferation [28]. These negative effects are all examined under “high concentration” or “overexpression” of OPN in vitro. In addition, numerous studies have focused on the negative effects of phosphorylated OPN on HA formation and growth due to its higher affinity for the HA crystals [25] as well as the positive effects on osteoclast adhesion [29,30] and bone resorption [16]. However, our study using a “super high concentration” (more than 500,000 times higher than physiological level in mice) of rOPN solution without phosphorylation treatment showed the positive effect on osteogenesis. This phenomenon can be explained by the fact that unphosphorylated OPN has no HA-inhibiting activity up to a certain high concentration [25]. Katayama et al. reported that unmodified recombinant-rat-OPN (rrOPN) promoted the actin ring formation of osteoclasts and the attachment of osteoblast-like cell lines on the culture dish in addition to the decreased effect on mediating the osteoclast attachment compared with phosphorylated rrOPN; unphosphorylated rrOPN also could mediate the attachment of osteoclasts [29]. Consequently, the concentration of rOPN around the implant could be diluted in our in vivo experiment model by diffusing into the gap between implant and pre-existing bone, resulting in enhancing the direct osteogenesis. Challenges by changing the concentration of rOPN solution are expected to increase the positive effect on direct osteogenesis. Further investigation is needed to find a proper concentration for direct osteogenesis and determine whether such dephosphorylated OPN has regulatory significance in the in vivo situation.

The rOPN group represented a significantly increased OPN-positive perimeter rate around the implant surface compared with the RGDS and HABP-OPNpep groups, while there were no significant differences between the OPN pep group in week 2. These results are consistent with studies finding that RGD-containing peptides inhibit osteoclasts from adhering to the OPN [31,32]. The synthetic RGDS peptide also prevents osteoclast-like multinucleated cells forming actin rings in a dose-dependent manner [17]. The observed OPN-positive reaction on the implant surface in this study was considered to be mainly deposited by osteoclast-lineage cells migrating around the implant [9], since OPN immunoreaction was negative around the implant surface on day 1 in the rOPN group in the WT mice (data not shown). It is understandable that RGDS peptide inhibits the OPN secretion by osteoclasts on the implant surface by blocking the RGD-recognizing receptors (integrins) in the cell surface, whereas rOPN promotes osteoclasts to secrete OPN on the implant surface resulting in the increased rate of OPN deposition followed by direct osteogenesis. Since OPN also affects osteoblast migration, adhesion, and differentiation to form the bone matrix, other regions of amino acid sequence of the OPN may contribute to affect osteoclast activity including OPN secretion. In contrast, the RDG motif in the OPN protein did not contribute to the activation of osteogenic cells to form the bone matrix. Consistent with these concepts, the synthetic peptides based on OPN sequences such as OPNpep and HABP-OPNpep as well as RGDS peptide could be potent inhibitors for direct osteogenesis by blocking the receptors of osteogenic cells. Thus, it is possible that the HABP promoted the disturbing effect of OPNpep on the implant surface. At the initial trial, our choice of peptides was based on the idea that synthetic peptides can be less expensive than rOPN protein and we picked its symbolic RGD motif including the adjacent sequence SLAYGLR which serves as a cryptic binding site for additional integrins [33,34,35] from rOPN. For making maximum impact with minimum cost, further studies are needed to identify minimum amino acid sequences from the rOPN protein that contribute to activation of osteogenic cells on the implant surface leading to faster osseointegration.

Any protein/peptide coating on the implant surface failed to rescue the healing events in the *Opn*-KO mice resulting in little direct osteogenesis. The healing patterns at the bone-implant interface were almost the same as those in our previous study using HAB-implant without protein/peptide coating in the *Opn*-KO mice [9]. OPN is also known as a pleiotropic cytokine and its expression is up-regulated during inflammation. For example, T cells express OPN rapidly after activation, suggesting that this protein is associated with immune reaction and host defense [36]. Several publications also reported that OPN is an important regulator involved in inflammatory responses, immune cell function, tissue reconstruction, and vascular remodeling [37,38,39]. Additionally, O’Regan et al. defined a role for OPN in regulating inflammatory cell accumulation and function at sites of inflammation and tissue repair [40]. Osseous wound healing around a dental implant in mice is distinguished into four phases based on the histological findings: inflammatory, proliferative, formative, and remodeling phases [9,11]. Histological sections of the *Opn*-KO mice showed long retention of the inflammatory phase in the healing process after the implantation. Thus, *Opn* deficiency affects the healing process for achieving osseointegration due to the defective immune system followed by the disturbance of proliferative and formative phases, irrespective of the addition of exogenous OPN in the *Opn*-KO mice.

Notably, there was a significant increase in proliferative activity of the HABP-OPNpep group in the early days after the implantation in the WT mice, whereas no increases were found in the OPNpep group without HABP in this study. Possible explanation is that a peptide with both RGD and SLAYGLR motifs derived from rOPN might stimulate the proliferative activity of cells around the implant in the early stage of the healing process after implantation and HABP might enhance OPNpep to bind to the implant surface, leading to continuous supply of OPNpep to cells around its surface. A previous study has reported that OPN promotes bone regeneration by inducing stem cell proliferation and by enhancing angiogenic properties [41]. In contrast, the clear effect on cell proliferation was not observed in the rOPN group which showed promotion of direct osteogenesis finally in this study. Moreover, the HABP-OPNpep group did not show any progress in direct osteogenesis. According to the healing process after implantation, the proliferation phase is characterized by changing the number of proliferative cells to end this phase by decreasing proliferative cells followed by cell migration and differentiation phases. Considering these results, higher activity of cell proliferation disturbs the next phase of healing progression. Therefore, moderate effects on cell proliferation may be favorable for final osseointegration. Thus, certain amino acid sequences from rOPN that are involved in cell proliferation, migration, differentiation, bone formation, and remodeling should be identified to find the best combination of those sequences for maximum effects on direct osteogenesis. This is an area for future study.

## 4. Materials and Methods

### 4.1. Animals and Experimental Procedure

All animal experiments were conducted in compliance with ARRIVE guidelines and a protocol that was reviewed by the Institutional Animal Care and Use Committee and approved by the President of Niigata University (Permit Number: SA00783). *Opn*^-/-^ (B6.Cg-*Spp*1*^tm^*^1*Blh*^/J) and male WT (C57BL/6J: inbred strain of laboratory mouse) mice were purchased from Jackson Laboratories (Bar Harbor, ME, USA) and Charles River Laboratories Japan (Yokohama, Japan), respectively. We used 4 female and 2 male bred *Opn*-KO mice in this study. They were housed with a maximum of 5 mice per cage, with Palsoft (made from paper) for bedding purchased from Oriental Yeast Co, Ltd. (Tokyo, Japan), at around 23 °C and 50–70% humidity with food and water ad libitum on a 12 h light-dark cycle. A targeting vector containing the neomycin-resistant cassette and the *Herpes simplex virus thymidine kinase* gene was used to disrupt exons 4–7 of the *Opn* gene [42]. All surgeries were conducted under anesthesia using an intraperitoneal injection of a mixed solution (0.05–0.1 mL/10 g) of Domitor^®^ (1.875 mL: Nippon Zenyaku Kogyo Co, Ltd., Koriyama, Japan), midazolam (2 mL: Sandoz KK, Tokyo, Japan), Vetorphale^®^ (2.5 mL: Meiji Seika Pharma Co, Ltd., Tokyo, Japan), and physiological saline (18.625 mL).

### 4.2. Immediate Implant Placement

We extracted the upper-right first molar (M1) of four-week-old mice using a pair of dental forceps with modification (Figure 5a,b). Subsequently, a cavity was prepared on the alveolar socket of M1 using a drill (a trial piece: Kentec, Tokyo, Japan) with a gripper (SPI02: Kentec) (the diameter and depth of the cavity were 1.0 mm and <2.0 mm, respectively). We soaked the HAB-implants [11] in different OPN modified protein/peptides solution including rOPN protein (R&D Systems, Minneapolis, MN, USA; catalog no. 441-OP) (rOPN group; 20 μM in phosphate buffer saline [PBS]), Gly-Arg-Gly-Asp-Ser peptide (Peptide Institute Inc, Osaka, Japan; Fibronectin Active Fragment, #4189) (RGDS group; 3.1 mM in PBS), OPN mimic peptide (sequence: GRGDSLAYGLR [OPNpep group], theoretical molecular weight: 1164.27, purity: 97.95% [HPLC method], 3.1 mM in PBS), HABP-OPNpep (sequence: GGGLHAHKKPTQDIRGGGRGDSLAYGLR [HABP-OPNpep group], theoretical molecular weight: 3101.48, purity: 98.8% [HPLC method] 3.1 mM in PBS) (both peptides were provided from GenScript Japan [Tokyo, Japan] according to our order), and PBS (control) for 2 min in addition to filling the cavity with each solution before implant placement. The implant soaked in each solution was inserted into the cavity using a screwdriver (Prosper, Kashiwazaki, Japan) after controlling the bleeding from the extraction site (Figure 5c). We preferred the maxilla to the mandible in this study due to the stability and reproducibility of experiments in spite of the problem of initial implant stability. It is easy to fix the maxilla during operation, compared with the fixation of the mandible, resulting in the advantage for extraction of the molar as well as the implant placement. There were no adverse events or unexpected deaths as well as no significant differences in body weight between groups at any point, and all animals had general good health through the experimental periods.

### 4.3. Histological Procedure

Materials were collected from groups of the *Opn*-KO and WT mice at intervals of three, five days and one, two weeks after implantation (*n* = 96: Table 1). We used the same samples (*n* = 10) from the previous study as the control group to minimize the number of experimental animals [9]. At each interval, the animals were perfused with physiological saline transcardially followed by 4% paraformaldehyde in a 0.1 M phosphate buffer (pH 7.4) under deep anesthesia using an intraperitoneal injection of a mixed solution of Domitor^®^, midazolam, Vetorphale^®^, and physiological saline. The maxillae including the implants were removed *en bloc* and immersed in the same fixative for an additional 24 h. Following decalcification in a 10% ethylenediaminetetraacetic acid-2Na solution for 3 weeks at 4 °C, the specimens were dehydrated using an ethanol series and embedded in paraffin after removal of the implants, and 4 μm sagittal sections of the maxillae were prepared. The implant was carefully removed from the cavity using a screwdriver (Prosper) to minimize damage to the bone-implant interface. The paraffin sections were mounted on Matsunami adhesive silane (MAS)-coated glass (Matsunami Glass Ind., Osaka, Japan) slides, stained with hematoxylin and eosin (H&E), and processed for Azan-staining. Our observation focused on the lateral side of the bone-implant interface after removing the implant (Figure 5d).

### 4.4. Immunohistochemical and Histochemical Analysis

Immunohistochemistry using a rabbit anti-OPN polyclonal antibody diluted to 1:5000 (LMS Co., Ltd., Tokyo, Japan; catalog no. LSL-LB-4225) and a rat anti-Ki67 monoclonal antibody diluted to 1:100 for cell proliferation assay (Dako Japan, Tokyo, Japan; catalog no. M7249) was conducted with the Envision+/horseradish peroxidase system (Dako, Tokyo, Japan; catalog no. K5027) and the avidin-biotin peroxidase complex (Vectastain ABC Kit, Vector Laboratories) method with biotinylated anti-rat IgG (Vector Laboratories, CA, USA; catalog no. BA-4000), respectively. For final visualization of the sections, 0.05 M Tris-HCl buffer (pH 7.6) containing 0.04% 3-3′-diaminobenzidine tetrahydrochloride and 0.0002% H_2_O_2_ was used. The immunostained sections were counter-stained with H&E and 0.05% methylene blue. For control experiments, the primary antibodies were replaced with non-immune serum or PBS. For the histochemical demonstration of TRAP activity, the azo-dye method was utilized with slight modification [43].

### 4.5. Statistical Analysis

The number of Ki67-positive cells at the bone-implant interface of each specimen (208 × 159 μm^2^ grid was selected) was counted by the counter tool of Photoshop 2021 (Adobe Inc, San Jose, CA, USA). Data were obtained from 96 maxillae from the *Opn*-KO and WT mice (Table 1) for cell proliferation assay using the immunoreactivity of Ki67. The rate of OPN-positive perimeter around the implant or direct and indirect osteogenesis was statistically analyzed in the OPN immunostained or H&E stained sections using two-tailed Student’s *t*-test in the same way as our previous study [9]. Furthermore, the rate of osseointegration in the control group and the number of Ki67-positive cells among different stages after implantation was compared using one-way analysis of variance (ANOVA) followed by the Bonferroni test for multiple comparisons and the rates of osseointegration and OPN-positive perimeter or the number of Ki67-positive cells between different groups were compared using two-tailed Student’s *t*-test with statistical software after the confirmation of data normality and homogeneity of variance (SPSS 16.0J for Windows; SPSS Japan, Tokyo, Japan). The threshold for significance was defined (α = 0.05). The samples showing no normal distribution (OPN-perimeter at week 2: RGDS and HABP-OPNpep groups; rate of osseointegration at week 2: RGDS and HABP-OPNpep groups; Ki67 at day 5: HABP-OPNpep group) were compared by Kruskal–Wallis test followed by the Bonferroni test for multiple comparisons for more than three groups or Mann–Whitney U test for two groups. Data were reported as mean + SD, n indicated the sample number and p denoted the *p*-Value.

## 5. Conclusions

None of the implant surfaces could rescue the healing events in the *Opn*-KO mice due to their defective immune system. We found a significant increase in proliferative activity of the HABP-OPNpep group in the early days after implantation in the WT mice. A peptide with both RGD and SLAYGLR motifs derived from mouse OPN might stimulate proliferative activity of the cells around the implant in the early stage of the healing process resulting in the disturbance of direct osteogenesis. The rOPN group showed significantly increased rates of OPN-positive perimeter and direct osteogenesis around the implant surface compared to the RGDS group in week 2 in the WT mice. These results suggest that rOPN promotes OPN deposition on the implant surface, whereas the RGDS peptide inhibits this process. The rate of direct osteogenesis in week 4 in the control group was already achieved in week 2 in the rOPN group in the WT mice, suggesting that rOPN on the implant surface accelerates direct osteogenesis after implantation leading to its potential use in bone tissue engineering. Nevertheless, we have to consider the adverse effects of OPN on the body as well as concerns relating to other ECM proteins. OPN is involved in tumor progression such as cell proliferation, angiogenesis, and metastasis [44]. Further understanding of the implications of OPN in terms of adverse effects and its crosstalk with other ECM proteins could help develop better therapeutic strategies for the acceleration of direct osteogenesis.

## Figures and Tables

**Figure 1 ijms-23-01039-f001:**
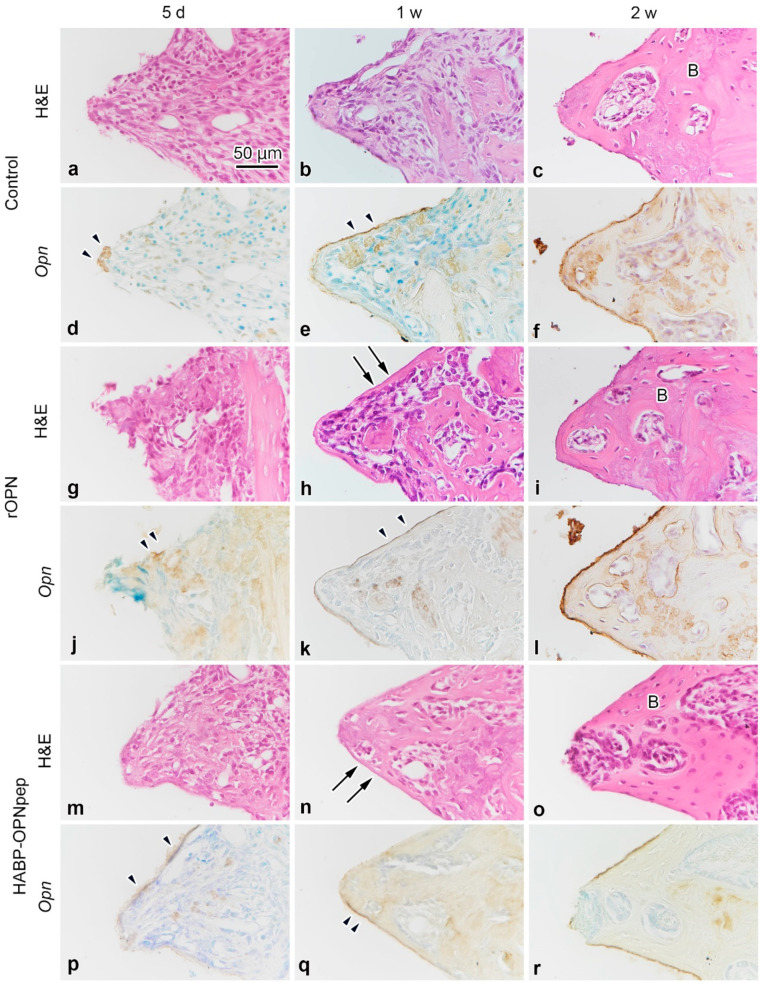
H&E-staining and OPN-immunoreactivity in the tissues surrounding the implants in the WT mice. (**a**,**g**,**m**) H&E-staining in the tissues surrounding the implants at day 5 after implant placement in the control, rOPN, and HABP-OPNpep groups. The infiltration of inflammatory cells and spindle-shaped or flattened cells is observed at the bone-implant interface. (**d**,**j**,**p**) There is a weak OPN positive immunoreaction at the bottom parts of threads and the cement lines of the pre-existing bone (arrowheads). (**b**,**c**,**h**,**i**,**n**,**o**) The formation of direct osteogenesis is clearly observed at week 1 (arrows) and week 2. (**e**,**f**,**k**,**l**,**q**,**r**) OPN-immunoreactivity gradually becomes intense (arrowheads) and elongates along the implant surface at week 2. B, bone. Scale bar = 50 μm.

**Figure 2 ijms-23-01039-f002:**
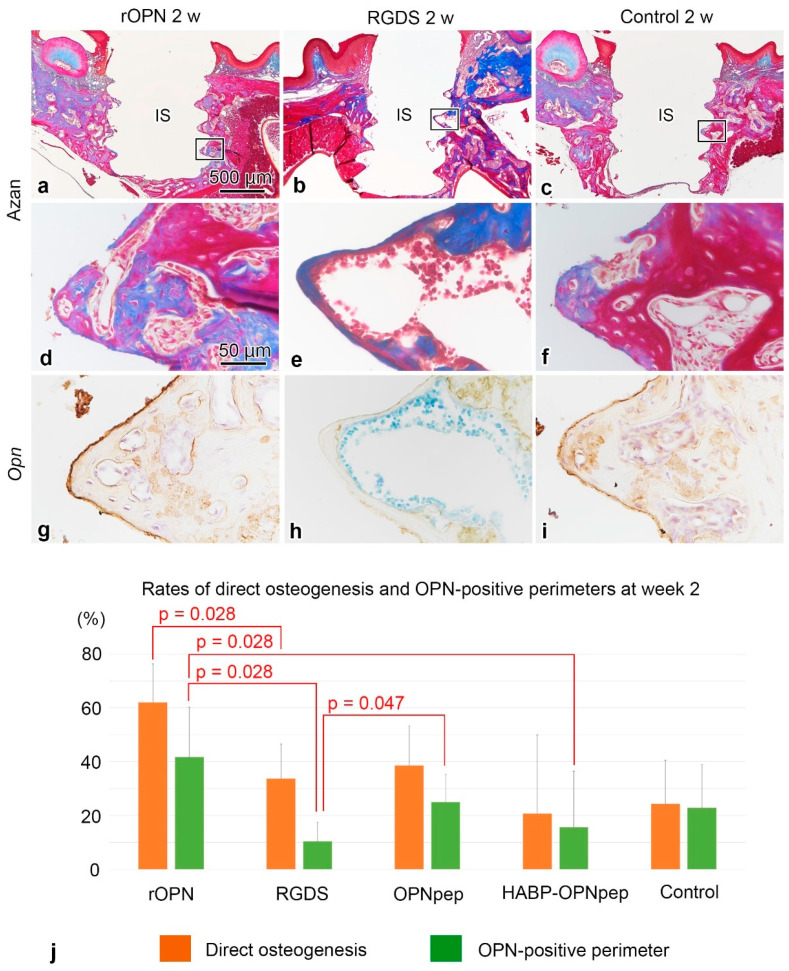
Azan-staining, OPN-immunoreactivity, and the rates of direct osteogenesis or OPN-positive perimeters in the WT mice. (**a**–**f**) Azan-staining in the tissues surrounding the implants at 2 weeks after implant placement in the rOPN, RGDS, and control groups. Partially, indirect osteogenesis progresses from the pre-existing bone in addition to the direct osteogenesis to achieve osseointegration at week 2. (**d**–**f**) are higher magnifications of the boxed areas in (**a**–**c**), respectively. (**g**–**i**) The OPN-immunoreactive lines coincide with the places where the direct osteogenesis occurs, although some areas lack the OPN reaction. (**j**) Quantification of the rates of direct osteogenesis and OPN-positive perimeters in rOPN (*n* = 5), RGDS (*n* = 5), OPNpep (*n* = 5), HABP-OPNpep (*n* = 5), and control (*n* = 3) groups. The RGDS and HABP-OPNpep groups without following normal distribution were compared by Kruskal–Wallis test followed by the Bonferroni test for multiple comparisons for more than three groups or Mann–Whitney U test for two groups. In the rOPN group, the OPN-positive perimeter around the implant surface significantly increases compared with that in the RGDS and HABP-OPNpep groups at week 2 and shows the highest rate compared with other groups. The OPN-positive perimeter in the OPNpep group is higher than that in the RGDS group. Statistical analysis used a two-tailed Student’s *t*-test or Mann–Whitney U test. As to the percentage of the vertical axis, the numerator is direct osteogenesis or OPN-positive perimeter around implant surface and the denominator is the perimeter of external surface of implant body. The rates are the mean + SD. IS, implant space. Scale bars = (**a**–**c**) 500, (**d**–**i**) 50 μm.

**Figure 3 ijms-23-01039-f003:**
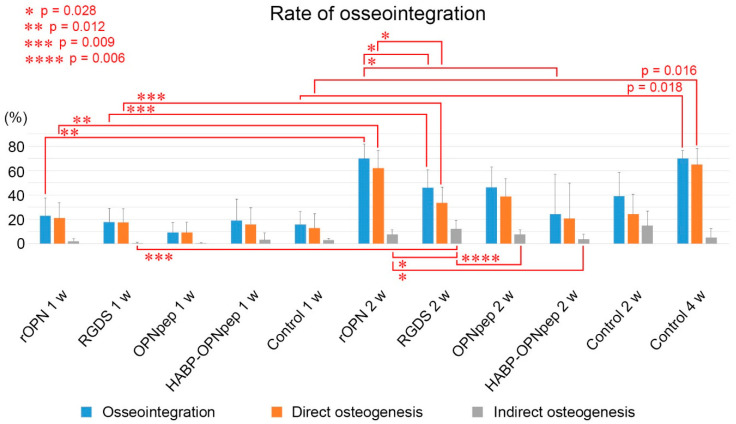
The rate of osseointegration that consists of direct and indirect osteogenesis in the WT mice. Quantification of the rates of osseointegration at week 1 in rOPN (*n* = 3), RGDS (*n* = 5), OPNpep (*n* = 5), HABP-OPNpep (*n* = 5), and control (*n* = 3) groups or week 2 in rOPN (*n* = 5), RGDS (*n* = 5), OPNpep (*n* = 5), HABP-OPNpep (*n* = 5), and control (*n* = 3) groups. The RGDS and HABP-OPNpep groups at week 2 without following normal distribution were compared by Kruskal–Wallis test followed by the Bonferroni test for multiple comparisons. The rate of osseointegration in the control group among different stages after implantation is compared using one-way analysis of variance (ANOVA) followed by the Bonferroni test for multiple comparisons. The rates of osseointegration between two groups are compared using two-tailed Student’s *t*-test. The rate of osseointegration in the rOPN group at week 2 is significantly higher than that in the RGDS group. Note that there were no significant differences in the rate of direct osteogenesis between the rOPN group at week 2 and the control group at week 4, showing that the direct osteogenesis ratio at week 4 in the control group is already achieved at week 2 in the rOPN group. The rates are the mean + SD.

**Figure 4 ijms-23-01039-f004:**
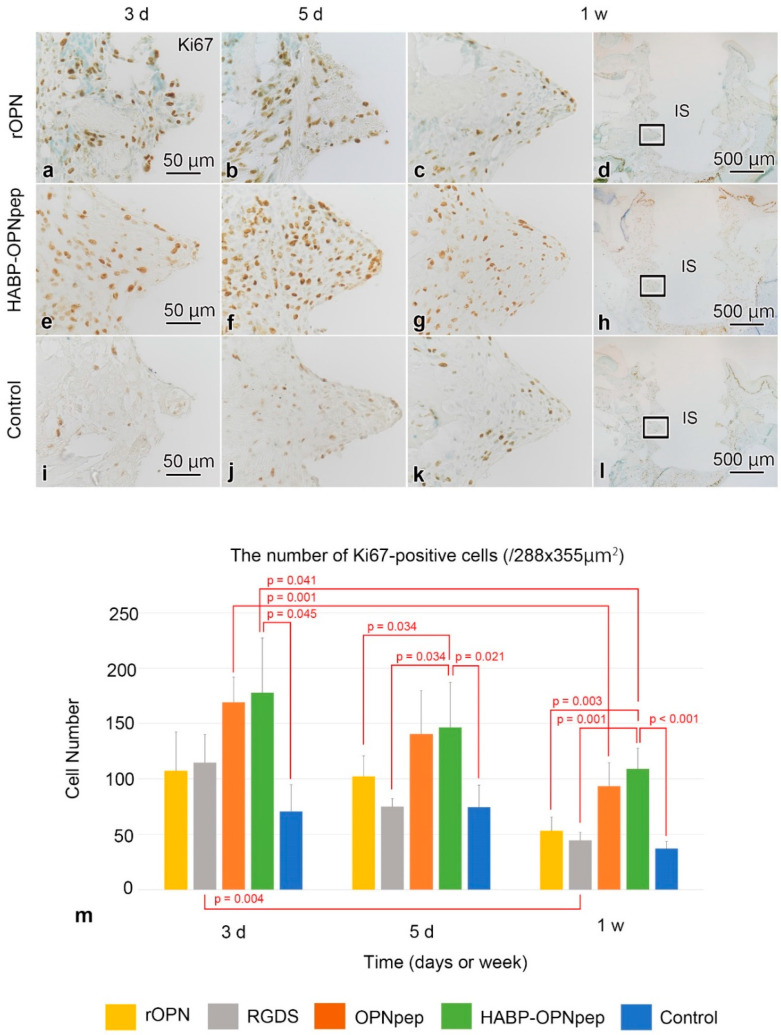
Ki67-immunoreactivities in the surrounding tissues and the number of cell proliferation in the WT mice. (**a**–**l**) Active cell proliferation tends to occur in the tissues surrounding the implant during days 3–5 and decreases at week 1 after implantation in the rOPN, HABP-OPNpep, and control groups. (**c**,**g**,**k**) are higher magnifications of the boxed areas in (**d**,**h**,**i**), respectively. (**m**) Quantification of the number of Ki67-positive cells at day 3 in rOPN (*n* = 3), RGDS (*n* = 5), OPNpep (*n* = 5), HABP-OPNpep (*n* = 4), and control (*n* = 3) groups or day 5 in rOPN (*n* = 4), RGDS (*n* = 3), OPNpep (*n* = 4), HABP-OPNpep (*n* = 4), and control (*n* = 4) or week 1 in rOPN (*n* = 3), RGDS (*n* = 5), OPNpep (*n* = 5), HABP-OPNpep (*n* = 5), and control (*n* = 3) groups. The number of Ki67-positive cells among different stages after implantation is compared using one-way analysis of variance (ANOVA) followed by the Bonferroni test for multiple comparisons and the number of Ki67-positive cells between two groups are compared using two-tailed Student’s *t*-test. The HABP-OPNpep group at day 5 without following normal distribution is compared by Kruskal–Wallis test followed by the Bonferroni test. The proliferative activity in the HABP-OPNpep group is significantly higher than that in the control group from day 3 to week 1 or those in rOPN and RGDS groups from day 5 to week 1. The numbers are the mean + SD. IS, implant space. Scale bars = (**d**,**h**,**l**) 500, (**a**–**c**,**e**–**g**,**i**–**k**) 50 μm.

**Figure 5 ijms-23-01039-f005:**
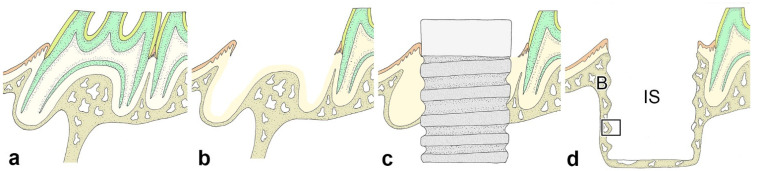
Schematic illustration showing the key steps of implant placement, including before (**a**) and after (**b**) tooth extraction of maxillary first molar, and protein/peptides coated HAB-implant placement after cavity preparation (**c**). Following the implant removal, the tissue surrounding the implant was observed (**d**). The black frame is the observation area in the histological section. B, bone; IS, implant space.

**Table 1 ijms-23-01039-t001:** Number of animals for histological and immunohistochemical analyses for Ki67 and OPN and TRAP histochemistry.

Strain	Method	Day 1	Day 3	Day 5	Week 1	Week 2	Week 4	Total
WT mice	Histological section	3	20 {3 ^1^}	19 {3 ^1^}	21	23	4 {4 ^1^}	90 {10 ^1^}
	Ki67	-	(20 {3 ^1^})	(19 {3 ^1^})	(21)	(23)	-	(86 {6 ^1^})
	OPN	(3)	(20 {3 ^1^})	(19 {3 ^1^})	(21)	(23)	-	(86 {6 ^1^})
	TRAP	-	(20 {3 ^1^})	(19 {3 ^1^})	(21)	-	-	(60 {6 ^1^})
*Opn*-KO mice	Histological section	-	-	-	-	6	-	6
	OPN	-	-	-	-	(6)	-	(6)
Total		3	20 {3 ^1^}	19 {3 ^1^}	21	29	4 {4 ^1^}	96 {10 ^1^}

^1^ The samples used in the previous study [9].

## Data Availability

All data generated or analyzed during this study are included in this published article and its Appendix A.

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
