# Peer review of "Osteopontin on the Dental Implant Surface Promotes Direct Osteogenesis in Osseointegration"

_ijms, 2022, doi:10.3390/ijms23031039_

Round 1
Reviewer 1 Report
Dear authors, I don't want to question the scientific nature of your research, but I believe that it is necessary to make some considerations:
Why you have examined the OPN, which is known to have various functions, including adverse ones for the body (for example, it is expressed in various cancers such as of lung, breast, colon, ovarian cancer etc) and it is involved in the formation of kidney stones and also responsible for one of the mechanisms of metastasis in breast cancer)?
You could have done a study on other proteins more directly involved in osteogenesis processes such as Osteocalcin, for example.
I believe that this research has not clinical significance and above all it does not reveal certain but only supposed mechanisms of action.
Author Response
We appreciate your critical evaluation. Our previous study has demonstrated that direct osteogenesis on the dental implant surface is significantly disturbed in the Opn-knockout (KO) mice. This is one of the reasons why we have examined the OPN in the process of osseointegration after dental implantation. Against the loss-of-function of OPN in the experimental model using the Opn-KO mice in our previous study, we applied the gain-of-function of OPN using the OPN modified protein/peptides coating implants to analyze their effects on the healing patterns of the bone-implant interface after immediately placed implantation in the maxillae of 4-week-old Opn-KO and wild-type mice. Nevertheless, we have to consider the adverse effects of OPN on the body as well as the concerns of the other ECM proteins as the reviewer points out. OPN is involved in tumor progression such as cell proliferation, angiogenesis, and metastasis. Consequently, we added these important issues in the section of “Conclusions”: “Nevertheless, we have to consider the adverse effects of OPN on the body as well as the concerns of the other ECM proteins. OPN is involved in tumor progression such as cell proliferation, angiogenesis, and metastasis [49]. Further understanding of the im-plications of OPN in adverse effects and its crosstalk with the other ECM proteins could help develop better therapeutic strategies for the acceleration of the direct osteogenesis (Lines 411-416).”
Reviewer 2 Report
This is well designed and performed study. I don't have any comment on this study. It can be published as it is.
Author Response
We appreciate your positive evaluation.
Reviewer 3 Report
I suggest to reject this article!
Author Response
We appreciate your critical evaluation. We could improve the manuscript if you give us the concreate suggestions.
Reviewer 4 Report
Kudos to the author who conducted and planned the research.
I would like to suggest a few things that should be improved.
1. Please include the total number of animals in your abstract.
2. There are parts where the abbreviation is not defined. (ex. RDG) Abbreviations that appear for the first time in the abstract or text should be clearly defined.
3. Since the sentences related to the research results described in the last paragraph of the introduction are redundant, it is better to delete them. It is recommended to close the introduction with a clear presentation of the research purpose.
4. I am curious as to why the implant was placed in the upper jaw, where the initial fixation was unstable, instead of placing the implant in the mandible. If you have a reference or author's opinion, it's better to add it to the discussion section.
5. In figure 2, it is recommended to delete the horizontal bar without scale.
Author Response
Thank you for your kind suggestions. The points of improvement are as follows.
- We added the total number of animals in the section of “Abstract” according to the reviewer’s suggestion.
- We spelled out the abbreviation that appear for the first time in the abstract or text according to the reviewer’s suggestion.
- We improved the last paragraph of the “Introduction” according to the reviewer’s suggestion.
- We understand the reviewer’s notion. We preferred the maxilla to the mandible due to the stability and reproducibility of experiments in spite of the problem of initial implant fixation. It is easy to fix the maxilla during operation, compared with the fixation of the mandible, resulting in the advantage for extraction of the molar as well as the implant placement. We added this issue in the section of “Materials and Methods” according to the reviewer’s suggestion: “We preferred the maxilla to the mandible in this study due to the stability and reproducibility of experiments in spite of the problem of initial implant stability. It is easy to fix the maxilla during operation, compared with the fixation of the mandible, resulting in the advantage for extraction of the molar as well as the implant placement (Lines 342-346).”
- We deleted the horizontal bar without scale in Figure 2 according to the reviewer’s suggestion.
Round 2
Reviewer 4 Report
Thanks for your improvement for the study.
My opinion for the manuscript is acceptance.